# The Relationship between Flourishing and Depression in Children in the U.S. Using a Socioecological Perspective

**DOI:** 10.3390/ijerph17218246

**Published:** 2020-11-08

**Authors:** Chang-Yong Jang, Eun-Hyung Cho, Yi-Sub Kwak, TaeEung Kim

**Affiliations:** 1Department of Sport Science, Korea Institute of Sport Science, Seoul 01794, Korea; jangcy529@kspo.or.kr (C.-Y.J.); ehcho@kspo.or.kr (E.-H.C.); 2Department of Physical Education, Dong-Eui University, Busan 47340, Korea; ysk2003@deu.ac.kr; 3Department of Epidemiology, University of California, Irvine, CA 92697, USA

**Keywords:** children, flourishing, pediatric depression, sociological factors, quality of life

## Abstract

Children’s flourishing is likely to be associated with achieving a positive mental and physical quality of life, which is considered as an important factor for helping children to overcome psychological adversity during the critical stage of emotional development. This study examined the relationships between children’s flourishing and childhood depression. This was a cross-sectional study using the 2011–2012 National Children’s Health Survey in the U.S. The conceptual framework that guided this study was a modified ecological system theory model. Multiple regressions were performed to investigate the associations between flourishing and pediatric depression, controlling for demographics, physical activity-related behaviors, family and environmental conditions. A total of 45,309 children (representing 33,293,823 children at the population level) were identified in this study (mean age: 13.63 years; female: 48.7%). Children’s childhood depression was highly related to direct parenting functions, individual needs and environmental availabilities and accessibilities from a socioecological perspective. This study revealed multiple dimensions of how sociological factors influence children’s flourishing and mental health. Parents’ involvement in children’s physical activities and family and social support are crucial for children’s flourishing and mental health status. More attention needs to be paid to provide children with family and social support to help them to overcome and reduce childhood depression.

## 1. Introduction

The concept of “flourishing” has been defined as a new expression of happiness and well-being [1,2], which is the combination of self-motivated achievement and emotional happiness within an individual’s environments. One’s flourishing can be obtained through positive relationships and engagements with positive emotions and a meaningful purpose and is achieved by overcoming individual adversity and hardship by committing to meaningful activities (e.g., religious mission and prayer activities) and valuable voluntary community services (e.g., church, the YMCA and senior centers). Children’s flourishing has been referred to as the improvement and maintenance of physical health and emotional well-being in later life [3,4]. Evidence has shown that children who flourished are more likely to improve and promote their health and well-being as adults compared to those who did not flourish in the longitudinal setting [4]. Therefore, as flourishing develops children’s mental health into healthy and stable conditions, it is one of the resources to achieve psychological well-being by providing a positive and meaningful environment [3].

Pediatric depression and its related symptoms play a significant role in children’s quality of life. Childhood depression is a critical factor in the development of children’s emotional and mental health [5,6,7]. In a nationally representative survey, nearly 40% of children were shown to experience pediatric depression in the United States [8]. To decrease the depressive outcomes of children, the following factors should also be considered as risk factors of childhood depression: a family history of depression [7,9,10]; family dysfunction or caregiver–child conflict; [3,11,12] exposure to early adversity including abuse, neglect and early loss [3,13,14]; psychosocial stressors such as bullying, peer problems and academic difficulties [15]; a negative style of interpreting events and coping with stress [16]; a history of anxiety disorders [10,11]; and substance abuse [15]. In general, younger children who have experienced disappointing life events, parental abuse, family conflicts and/or psychopathology are more likely to experience pediatric depression [9,17].

In addition, research is needed regarding the recovery from depression. Without consistent care and observation, the recurrence of children’s depression cannot be ruled out. Individual factors of children affecting pediatric depression may include physical inactivity, a healthy, normal social life (e.g., school, community and church), obesity and the degree of sleep [3,13,18]. Psychosocial risk factors associated with these recurrences of pediatric major depression include the following: a history of prior depressive episodes, environmental stressors, limited social support, a family history of recurrent unipolar major depression or other psychopathology, presence of comorbid disorder and presence of residual depressive symptoms [18].

Furthermore, pediatric depression is highly associated with socioecological and hierarchical settings. Children have multiple roles in varied settings including the home and school, which are comprised of distinct characteristics. To better understand the influencing factors on children’s pediatric depression, a socioecological multi-dimensional approach should be considered.

It is proposed that children’s flourishing can play a key role in constructing the meaning of objective goods and motivating children to pursue a better life [1]. Therefore, children’s flourishing is likely to be associated with achieving a positive mental and physical quality of life. Flourishing may also be a key element in helping children to overcome psychological hardships during critical stages of their emotional development.

Moreover, children are less independent than adults, as they are nested in parental, family and societal environments such as their school and community. Therefore, healthy and sound environments as well as parental health, the safety of their neighborhood and school and the accessibility of facilities will play an important role in helping children to maintain a healthy weight and avoid pediatric depression. This study takes a multi-level research approach and presents a modified social–ecological system model-driven framework to account for dynamic inferences in a hierarchical structure [19]. Based on the model (see Figure 1), children are shown to experience many key transitions in their lives. The following three parts were incorporated into the framework: (1) children’s individual needs (e.g., health, emotional and behavioral development, family relationships etc.), (2) parenting capacity (e.g., level of education and income, etc.) and (3) family and environmental factors (e.g., family’s social integration, family structure and family history and functioning, etc.).

Consequently, it is imperative to investigate how children’s flourishing is associated with children’s depression and comprehensive socioecological-related factors in a large, non-clinical sample. Children’s flourishing is known as an active and positive motivational theory that has evolved to supersede conventional motivations. Therefore, we aim to determine how active and positive feelings and the levels of happiness of individual children affect pediatric depression. Additionally, this study examined several potential moderating factors including age, sex, race/ethnicity differences, being overweight and bullying. Specifically, it is hypothesized that children’s flourishing and pediatric depression are moderated by age, sex and ethnicity after controlling for socioecological factors.

More specific models used to test our hypotheses are listed as follows:Model 1: Children’s flourishing affects their depression when children’s individual needs are controlled (e.g., health, emotional and behavioral development, family relationships, etc.);Model 2: Children’s flourishing affects their depression when individual needs and parenting capacities are controlled (e.g., level of education and income etc.);Model 3: Children’s flourishing affects their depression when individual needs, parenting capacities and the family are controlled (e.g., family’s social integration, family history and functioning etc.);Model 4: Children’s flourishing affects their depression when individual needs, parenting capacities and family and environmental factors are controlled (e.g., family structure etc.).

## 2. Materials and Methods 

### 2.1. Study Population and Sampling

This study was a cross-sectional study; the data for this study were extracted from the 2011–2012 National Survey of Children’s Health (NSCH) in the United States. The NSCH employed random sampling and telephone household interviews conducted by the National Center for Health Statistics of the Centers for Disease Control and Prevention [20]. Participants of the NSCH were non-institutionalized children aged 0–17 years, weighted according to the population nationally, and the survey included information on a variety of physical, emotional and behavioral health indicators relating to children and their family, neighborhoods and communities. For this study, since some information about respondents aged younger than 10 years could not be obtained or was limited, we extracted data from children aged 10–17 years. A total of 45,309 children (representing 33,293,823 children at the population level) were recruited for this study (mean age: 13.63, SD: 2.35 years; female: 48.72%). This study was approved and exempted by the Institutional Review Board at its institution due to the use of secondary data. 

### 2.2. Measurement 

Pediatric depression was assessed using a four-point Likert scale (ranging from 1 to 4) including questions such as “Would you describe your child’s depression as mild, moderate, or severe?”. A composite measure for flourishing was measured using a five-point Likert scale (ranging from 1 to 5), and the scale’s internal consistency was determined to be good for the current study (α = 0.75; the generally acceptable reliability value in social science is α = 0.70 [21]). Examples of questions were as follows: (a) “(He⋅She) shows interest and curiosity in learning new things”, (b) “(He⋅She) stays calm and in control when faced with a challenge” and (c) “(He⋅She) finishes the tasks (he⋅she) starts and follows through with what (he⋅she) says (he will⋅she will) do”. 

Eight variables corresponding to children’s demographic and behavioral indicators were collected: age, sex, race/ethnicity, sedentary behavior, after-school activities, physical activity, days of school missed and quality of sleep. The range of ages was 10–17 years old. Four groups of participant’s race/ethnicity were categorized: non-Hispanic White, non-Hispanic Black, Hispanic and other. A composite scale for sedentary behavior was computed as the product of the unit of measurement (hours) and the length of activity time for activities such as playing video games and/or watching TV, videos or DVDs. For after-school activity, three binary scales including taking sports classes or being on a sports team, participation in any clubs/organizations and taking part in any other organized activity during the past year calculated a composite measure. Physical activity was assessed by how many days children exercised, played a sport or participated in vigorous physical activity for at least 20 min. The number of days missed because of illness or injury during the past year was measured for school attendance. Quality of sleep was assessed by how many nights the participant slept well during the previous week. 

Parenting capacity for sociological factors related to pediatric depression included parents’ health and ability to cope with stress, smoking cigarettes, alcohol consumption and drug use, parents’ involvement and having a mentor. Parents’ health was assessed as the average of the mothers’ and fathers’ physical and pediatric depression. Parents’ ability to cope with stress was measured with a four-point Likert scale that measured how well parents felt they were coping with the daily demands of their children. Two binary items for parents’ health behaviors were employed to assess cigarette smoking, alcohol consumption and drug use. A summation of three items for parents’ involvement measured how often their children were involved in any type of community service or volunteer work at school, church or in the community during the past year; attending events or activities that their children participated in; and how many of their children’s friends they have met. Lastly, a binary item for having mentors was used to determine whether parents had a mentor in the school, neighborhood or community on whom they could rely for guidance. 

Four variables of family function related to pediatric depression were collected: the parent and child relationship, family activity, number of family members aged younger than 18 years and federal poverty level (FPL). The parent and child relationship was measured with a four-point Likert scale asking how well a parent shared ideas or talked about things that really matter. Two scales including how often children attended a religious service and how many days during the past week that all the family members came together to eat were assessed as a composite for family activity. A four-point Likert-scale item assessed the number of family members aged younger than 18 years. A participant’s parent level of poverty was categorized into four groups based on the FPL: 0–99% FPL, 100–199% FPL, 200–399 FLP and ≥400% FPL [22].

Environmental factors related to children’s depression consisted of a single parent variable, social capital, neighborhood amenities and perceived safety around their current residence. Initially, nine categories of combined family structure and the marital/cohabitation status of a child’s parent(s) were presented; however, binary variables, such as single-parent vs. another family structure, were restructured due to single-parent children’s pediatric depression statuses possibly differing from those from a different family structure. A composite scale for social capital was computed that asked four continuous items: the extent to which respondents agreed that (a) they watched out for each other’s children, (b) people helped each other out, (c) there were people they could count on and (d) if their children were outside playing and something happened, there were adults nearby who they could trust to help their child. A summation of three binary items for neighborhood amenities was calculated including whether sidewalks or walking paths, park or playground areas and a recreation or community center existed in their area. Lastly, perceived safety around their living conditions was measured as a composite of two items: how often they feel their child was safe in their community or neighborhood and at school.

### 2.3. Data Analyses

Unweighted and weighted descriptive statistics of the population were reported to describe the study participants’ demographic characteristics such as age, sex, race and weight using means, standard deviations or percentages. A multiple regression was performed to examine the associations between socio-demographic variables (e.g., family activity and parents’ health behavior) and depression. In addition, the product of coefficients procedure [23] was used to determine if age, sex and race/ethnicity moderated any of these effects in each model. All statistical analyses were conducted using Stata^®^ 15.1, and statistical tests were conducted using a 0.05 alpha level and a 95% confidence interval.

## 3. Results 

The unweighted (e.g., number of participants) and weighted (e.g., mean, SD and percentage) descriptive statistics of the study population are shown in Table 1. In total, 45,309 children (of 33,293,823 children at the population level) were identified (mean age = 13.63 years; girls = 48.72%). As can be observed in Table 1, the participants’ ethnicity was quite diverse, with most of them being non-Hispanic White (53.89%) followed by Hispanic (20.77%) and non-Hispanic Black (14.00%). Their income levels were also diverse (see Table 1). For example, 30.13% of respondents were in the group of more than 400% FPL followed by 200–399% FPL (28.88%), 100–199% FPL (21.12%) and 0–99% FPL (19.87%).

Testing the correlation coefficient between variables, we found that multicollinearity was not an issue (tolerance value for all > 0.1; variance inflation factor value < 10) [24]. All assumptions of ordinary least squares regression were met, such as normality, linearity, homoscedasticity and independent random errors. As seen in Table 2, all sequential regression models with covariates (e.g., individual, parent, family and environment factors) for depression were significant (*p* < 0.01) based on our conceptual framework. Overall, the final model was significant, *F*(44, 89) = 49.36, and accounted for about 24% (R^2^ = 0.24) of the variance in children’s depression; collectively, ten independent variables significantly predicted depression in socioecological factors.

More specifically, regarding children’s individual needs, four predictors including school days missed (*β* = 0.01, *p* < 0.01), quality of sleeping (*β* = −0.04, *p* < 0.01) and physical activity (*β* = −0.01, *p* < 0.05) were significant. Moreover, interestingly, as expected, flourishing (*β* = −0.25, *p* < 0.05) was a significant predictor of children’s depression. Regarding parental capacity resources, three were significant: parents’ health (*β* = −0.05, *p* < 0.01), ability to cope with stress (*β* = −0.13, *p* < 0.01) and alcohol/drug use (*β* = 0.14, *p* < 0.01). Regarding family function, two factors were significant: the parent and child relationship (*β* = −0.09, *p* < 0.01) and income level (over 400% FPL) (*β* = 1.00, *p* < 0.01). Lastly, regarding environmental factors, there were two significant predictors: being a single parent (*β* = −0.11, *p* < 0.05) and perceived safety (*β* = −0.03, *p* < 0.01). In addition, to determine if age, sex and race/ethnicity moderated any of the effects of being overweight, bullying and flourishing on depression, we used the product of coefficients procedure [23]. Specifically, we re-ran the regression model with age, sex and race/ethnicity and entered overweight, bullying and flourishing after controlling for other socioecological factors. Neither the effects nor any of the individual terms at this step were significant, indicating that age, sex and race/ethnicity did not act as moderators.

## 4. Discussion

In this study, the association between pediatric depression and children’s flourishing was examined using a socioecological and hierarchical framework, including the role of parents and the family, school and the community. We employed a socioecological multi-dimensional approach to better understand the influences on children’s pediatric depression. Specifically, we examined whether children’s flourishing affected their depression and whether age, sex, ethnicity, being overweight and bullying were associated with this relationship. As hypothesized, children’s flourishing was negatively associated with depression. Sex and race/ethnicity differences were not significant moderators of children’s depression.

Children’s flourishing was a significant predictor of children’s depression. This is because flourishing refers to happiness and emotional well-being [1,2]. Children who flourish are likely to display positive emotions, relationships, engagement, a meaningful purpose and accomplishments in accordance with overcoming one’s hardships, adversity and trauma by becoming involved in voluntary community services. Therefore, children’s flourishing seems to directly affect pediatric depression. This finding is consistent with a study by Sin and Lyubomirsky [25], which reviewed 51 positive psychology interventions, including positive feelings, positive behaviors or positive cognitions, and showed enhanced well-being and decreased depressive symptoms. However, no age, sex or race/ethnicity differences were found to be significant moderators of children’s flourishing as seen in children’s depression when controlling for socioecological factors, even though race/ethnic differences themselves have been identified as significant predictors. It is difficult to explain the combined effect of childhood depression, gender and race-related flourishing. In order to reduce depression in children, proactive flourishing must be developed regardless of age, gender or race, which will lead to children’s happiness and voluntary achievement motivation.

Other vital individual risk factors in the development of depression, such as being overweight [26] and being involved in bullying, were hypothesized and examined [27]. None of them significantly accounted for childhood risk of depression. In addition, no age, sex or ethnicity differences in being overweight or bullying were revealed to be associated with children’s depression when controlling for socioecological factors. One possible explanation for this is that the data used for this study were obtained through random sampling and household telephone interviews by parents and/or caregivers. Parents/caregivers may not have known if their child was being bullied at school or in the community. Many studies have directly or indirectly found effects of childhood obesity on children’s depression [27,28,29]; however, in this study, we found that the degree of obesity in children did not affect depression in children. It is considered that the degree of depression in a child is difficult for parents to make a concrete and accurate judgment about, and the data did not reach a meaningful conclusion. However, childhood depression is a complex manifestation of the child’s state of body and mind [30], and further research is needed on flourishing. 

Parents’ risky behaviors (i.e., consumption of alcohol and drugs) were positively associated with depression in children. This is consistent with a previous study that showed that parents’ or guardians’ substance abuse can cause their children to suffer depression [15]. Parents having a good health status, as well as better stress-coping skills, also showed a negative relationship with signs of depression in children, which was confirmed by a previous study [15,16]. However, the importance of parental involvement in children’s depression was not significant, suggesting that parents choosing to volunteer at their children’s school, church or in the community would not lower their children’s risk of developing depression. Moreover, negative parental health behaviors including smoking were shown to cause depression in their children; however, parents’ smoking was not a significant factor in this study. In summary, parenting plays a key role in lowering rates of depression in children.

Regarding family function, children with a better relationship with their parents showed a decreased risk of depression, which was in line with other studies [3,11,12], and family dysfunction or caregiver-child conflict was found to negatively affect the development of depression in children. Other factors including family activity, number of family members and level of poverty were not highly associated with children’s depression. Although the family is a key component of children’s risk of depression [15], these variables did not significantly contribute in this study. However, research attention needs to be devoted to the positive influences of family functioning on children’s emotional well-being through better relationships with their parents, meaningful family activities, parent–child communication, family connectedness and interaction with family members [31,32].

Regarding children’s depression, environmental factors should also be considered. Children with a single parent were less likely to be depressed than those with two parents. Possible explanations for this include the idea that single parents may pay more attention to their children and try to help motivate their children more and encourage participation in diverse activities. Children living in a safe neighborhood and who attended a safe school were less likely to be depressed, perhaps because feeling comfortable with interacting in their community dampens the risk of depression in children. Other depression-related environmental factors such as social capital and neighborhood amenities were not significantly associated with children’s depression in this study. However, environmental stressors and limited social support are vital environmental confounders that could affect children’s risk of depression [18].

Finally, some key individual factors affecting children’s depression were also reported, including missed school days, physical inactivity level and sleep quality. The latter two were significant negative predictors of children’s depression. This is because poor sleeping habits negatively influence children’s emotional/psychological well-being and development [33,34,35,36]. On the other hand, children who have missed school more often were more likely to be depressed. Notably, school is not only important in terms of academic achievement but also as it allows children to become involved in social and emotional relationships with peers and teachers [36].

According to the explained variance (R^2^) for each model, which is the proportion of variance in the criterion variable that can be explained by the combined predictor variables, our results showed that a child’s individual variables (Model 1) in this study showed an explanatory power for depression in children with a proportion of about 20%. Remarkably, it was found that the proportion of the explanatory power of children’s depression increased as the parenting capacity (Model 2; 23%), family function (Model 3; 24%) and environmental characteristics (Model 4; 24%) were added to the individual factors of children. This directly or indirectly demonstrates that children need government or policy help in terms of the role and support of parents and families and environmental aspects that can improve children’s depression in order to reduce their depression and improve their quality of life.

In summary, from a socioecological perspective, children’s depression was highly associated with direct parenting functions, school, environmental factors and individual needs. Therefore, the role of parents in preventing and/or minimizing a child’s risk of developing depression was confirmed. Parents and teachers are role models to children [1] as children spend most of their time with one of these groups. As discussed, therefore, children’s flourishing may be a primary resource for helping children to overcome psychological adversities in their lives.

This study examined the relationships between flourishing and pediatric depression in children with several potential moderating factors including age, sex, race/ethnicity differences, being overweight and bullying. The outcomes improve our understanding of the individual and environmental factors that determine children’s psychological flourishing, which could lead to public health prevention and intervention strategies that develop children’s character and increase their quality of life. The rationale for this research was that once parents, family members, schools and the community at large are educated on children’s flourishing patterns, they can use this knowledge to prevent and minimize the rates of obesity and/or pediatric depression. This will also decrease the economic and social burdens on the community.

However, this study should be considered in light of the following limitations. First, all study variables were from respondents’ one-day recollection sand self-reported measurements, which may lead to inaccurate results due to recall bias or interview bias. Second, the generalizability of the results to the pediatric depression population is limited and cannot be extended to those living in other, more-diversely populated areas because of the small sample size. Therefore, a larger sample with more diverse participants would yield results that are more robust. Third, the dataset excluded households with no telephone, which may lead to a biased survey population because of the underrepresentation of certain participants. Therefore, future studies regarding the prevention of pediatric depression should consider these limitations.

## Figures and Tables

**Figure 1 ijerph-17-08246-f001:**
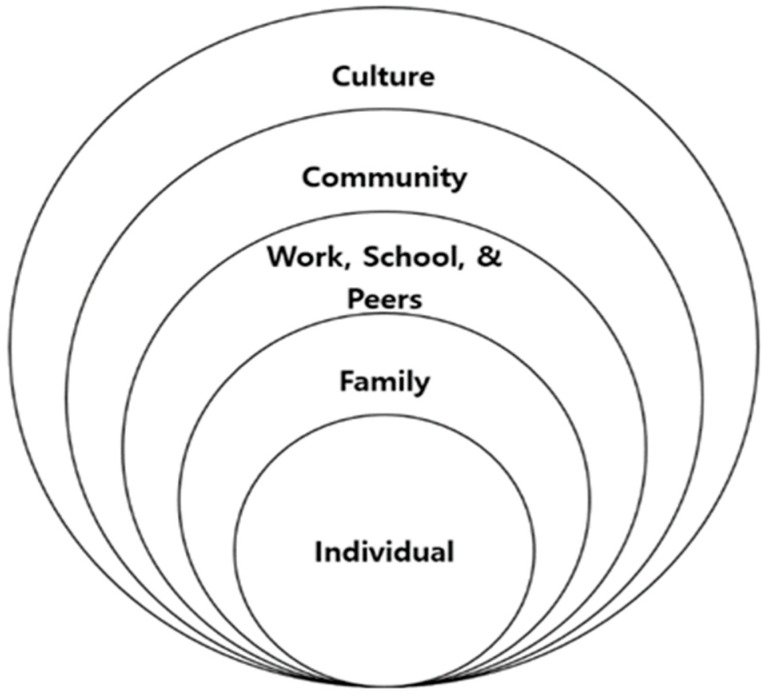
Modified socioecological system theory (Adapted form Bronfenbrenner [19]).

**Table 1 ijerph-17-08246-t001:** Descriptive statistics of the study sample.

Variables	% (*N*), Mean (SD)
Depression	1.87 (0.90)
1. Individual	
Flourishing	4.15 (0.69)
Age	13.53 (2.35)
Female	48.72% (21,658)
Race/ethnicity:	
Non-Hispanic white	53.89% (30,496)
Non-Hispanic black	14.00% (4242)
Hispanic	20.77% (5216)
Other	9.01% (4368)
After-school activity (days)	1.56 (0.10)
School missing (days)	3.85 (5.49)
Quality of sleeping (days)	5.81 (1.78)
Physical activity (days)	4.16 (2.32)
Sedentary behaviors (hours)	4.04 (3.51)
Overweight	29.84% (12,788)
2. Parenting capacity	
Parent health (degrees)	6.75 (2.40)
Parent involvements (frequencies)	8.51 (2.38)
Parent stress coping (degrees)	3.53 (0.58)
Smoking	24.51% (10,580)
Alcohol drugs	14.32% (6247)
Having mentors	83.37% (41,505)
3. Family function	
Family activity (frequencies)	6.84 (2.56)
Parent and child relationship (degrees)	3.62 (0.61)
Number of family members less than 18 years old (counts)	2.27 (0.99)
Federal poverty level (FPL):	
0–99% FPL	19.87% (6013)
100–199% FPL	21.12% (7747)
200–399% FPL	28.88% (14,052)
400% FPL	30.13% (17,497)
4. Environments characteristics	
Combined family structure & marital/cohabitation status:	
Single-mother	19.44% (7353)
Others	79.26% (37,416)
Social capitals(degree)	13.50 (2.94)
Neighborhood amenities (frequencies)	2.28 (0.95)
Perceived safety (frequencies)	6.86 (1.34)

*N* = 45,309, Weighted *N* = 33,293,823, Data source: 2011–2012 National Survey of Children’s Health.

**Table 2 ijerph-17-08246-t002:** Sequential regression analysis of children’s depression with covariates.

	Model 1		Model 2		Model 3		Model 4	
Covariates	Coef	95% CI	Coef	95% CI	Coef	95% CI	Coef	95% CI
1. Individual								
Flourishing	−0.39 **	(−0.63, −0.15)	−0.28 **	(−0.53, −0.03)	−0.27 *	(−0.51, −0.02)	−0.25 *	(−0.499, −0.001)
Flourishing * age	0.00	(−0.02, 0.02)	−0.00	(−0.02, 0.02)	−0.00	(−0.02, 0.02)	−0.00	(−0.02, 0.02)
Flourishing * female	−0.03	(−1.00, 0.04)	−0.01	(−0.09, 0.06)	−0.02	(−0.09, 0.06)	−0.02	(−0.09, 0.06)
Flourishing * Race/ethnicity:								
Non-Hispanic White	---		---		---		---	
Non-Hispanic Black	0.07	(−0.01, 0.16)	0.04	(−0.05, 0.12)	0.04	(−0.04, 0.13)	0.04	(−0.04, 0.13)
Other	0.03	(−0.49, 0.56)	−0.05	(−0.60, 0.49)	−1.00	(−0.64, 0.45)	−0.08	(−0.63, 0.46)
Age	0.01	(−0.080, 0.10)	0.02	(−0.07, 0.11)	0.02	(−0.07, 0.11)	0.019	(−0.07, 0.11)
Female	−29	(−0.060, 0.64)	0.21	(−0.15, 0.57,)	0.224	(−0.14, 0.59)	0.22	(−0.15, 0.59)
Race/ethnicity:								
Non-Hispanic white	---		---		---		---	
Non-Hispanic black	−0.51 *	(−0.91, −0.10)	−0.39	(−0.81, −0.03)	−0.39	(−0.80, 0.03)	−0.38	(−0.80, 0.04)
Hispanic	0.360	(−0.33, 1.05)	0.38	(−0.31, 1.08)	0.38	(−0.33, 1.09)	0.42	(−0.32, 1.12)
Other	0.03	(−0.49, 0.56)	−0.05	(−0.50, 0.49)	−1.0	(−0.64, 0.45)	−0.08	(−0.63, 0.46)
After school activity	0.01	(−0.01, 0.03)	0.03 *	(0.01, 0.05)	0.02	(−0.00, 0.04)	0.02	(−0.00, 0.04)
School missing	0.02 **	(0.01, 0.02)	0.01 **	(0.01, 0.02)	0.01 **	(0.01, 0.02)	0.01 **	(1.00, 0.02)
Quality of sleeping	−0.05 **	(−0.06, −0.03)	−0.04 **	(−0.05, −0.03)	−0.03 **	(−0.05, −0.02)	−0.04 **	(−0.05, −0.02)
Physical activity	−0.01 **	(−0.02, −0.01)	−0.01 *	(−0.02, −0.00)	−0.01 *	(−0.02, −0.00)	−0.01 *	(−0.02, −0.00)
Sedentary behavior	−0.00	(−0.01, 0.00)	−0.01	(−0.01, 0.00)	−0.01	(−0.01, 0.00)	−0.00	(−0.01, 0.00)
Overweight	0.14	(−0.09, 0.38)	0.13	(−0.10, 0.37)	0.15	(−0.09, 0.38)	0.15	(−0.09, 0.38)
Overweight * age	−0.00	(−0.02, 0.01)	−0.01	(−0.02, 0.01)	−0.01	(−0.02, 0.01)	−0.01	(−0.02, 0.01)
Overweight * female	0.04	(−0.04, 0.12)	0.03	(−0.05, 0.11)	0.03	(−0.05, 0.11)	0.03	(−0.05, 0.11)
Overweight * ethnicity:								
Non-Hispanic White	---		---		---		---	
Non-Hispanic Black	−0.08	(−0.18, 0.03)	−0.07	(−0.17, 0.04)	−0.07	(−0.18, 0.04)	−0.07	(−0.17, 0.04)
Hispanic	−0.04	(−0.16, 0.08)	−0.02	(−0.14, 0.10)	−0.01	(−0.13, 0.11)	−0.02	(−0.14, 1.00)
Other	−0.08	(−0.22, 0.07)	−0.04	(−0.19, 0.12)	−0.04	(−0.19, 0.12)	−0.04	(−0.20, 0.12)
Bullying	0.20 *	(0.03, 0.37)	0.17	(−0.01, 0.34)	0.17	(−0.00, 0.35)	0.16	(−0.00, 0.34)
Bullying * age	−0.00	(−0.02, 0.10)	−0.00	(−0.01, 0.01)	−0.00	(−0.02, 0.01)	−0.00	(−0.01, 0.01)
Bullying * female	−0.00	(−0.06, 0.05)	0.00	(−0.05, 0.06)	−0.00	(−0.06, 0.06)	−0.00	(−0.06, 0.06)
Bullying * ethnicity:								
Non-Hispanic White	---		---		---		---	
Non-Hispanic Black	0.07	(−0.00, 0.14)	0.07	(−0.01, 0.14)	0.07	(−0.01, 0.14)	0.07	(−0.01, 0.14)
Hispanic	0.01	(−0.08, 1.0)	0.02	(−0.08, 0.11)	0.02	(−0.08, 0.11)	0.02	(−0.07, 0.11)
Other	−0.01	(−0.09, 0.08)	−0.00	(−0.08, 0.08)	−0.00	(−0.08, 0.08)	−0.00	(−0.08, 0.08)
2. Parenting capacity								
Parents’ health			−0.04 **	(−0.05, −0.03)	−0.04 **	(−0.05, −0.03)	−0.05 **	(−0.07, −0.04)
Parents’ involvement			−0.01 *	(−0.02, −0.00)	−0.01	(−0.02, −00)	−0.01	(−0.02, −0.00)
Parents’ stress coping			−0.17 **	(−0.21, −0.13)	−0.14 **	(−0.18, −0.10)	−0.13 **	(−0.18, −0.09)
Smoking			0.02	(−0.02, 0.06)	0.03	(−0.01, 0.07)	0.02	(−0.02, 0.07)
Alcohol/drug use			0.13 **	(0.08, 0.19)	0.14 **	(0.08, 0.19)	0.14 **	(0.08, 0.19)
Having a mentor			−0.03	(−0.12, 0.06)	−0.03	(−0.12, 0.06)	−0.03	(−0.12, 0.06)
3. Family function								
Family activity					−0.01	(−0.02, 0.00)	−0.01	(−0.02, 0.00)
Parent and child relationship					−0.09 **	(−0.14, −0.05)	−0.09 **	(−0.13, −0.05)
No. family members aged <18 years					−0.01	(−0.03, 0.02)	−0.00	(−0.03, 0.02)
FPL								
0–99% FPL					--		--	
100–199% FPL					0.03	(−0.04, 0.09)	0.02	(−0.05, 0.08)
200–399% FPL					0.03	(−0.033,1.00)	0.03	(−0.04, 0.09)
≥400% FPL					1.00 *	(0.03, 0.17)	1.00 **	(0.03, 0.17)
4. Environmental characteristics								
Single parent							−0.11 *	(−0.18, −0.03)
Social capital							0.01	(−0.00, 0.01)
Neighborhood amenities							−0.01	(−0.03, 0.01)
Perceived safety							−0.03 **	(−0.05, −0.02)
R-squared	0.20		0.23		0.24		0.24	
Wald tests			F (6, 90) = 43.73 **		F (6, 90) = 6.71 **		F (4, 89) = 6.85 **	

Note: * *p* < 0.05; ** *p* < 0.01. Data source: 2011–2012 National Survey of Children’s Health. Weighted *N* = 33,293,823; Coef: Coefficient; CI: confidence interval; FPL: federal poverty level.

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
