# Peer review of "The Relationship between Flourishing and Depression in Children in the U.S. Using a Socioecological Perspective"

_ijerph, 2020, doi:10.3390/ijerph17218246_

Round 1

Reviewer 1 Report

Overall this second submission is improved and deals with issues noted in the initial review. The improvement in the results section is notable. There continue to be English language issues including but not limited to:

Line 39: "mission and praying" is not clearly stated, please revise

Line 43 revise: those who are not

Line 44 use active voice

Line 47 use active voice

Line 70 verb tense question- present or past tense?

Line 258 not clearly stated

Author Response

Thank you for your valuable review. We really appreciate the great comments and suggestions for the better research. Based on your comments, all changes are provided by a point-by-point response and also rewritten with red color in the manuscript. Please see the attachment.

Reviewer 2 Report

Dear authors,

Thank you for revising the manuscript. Following are my comments to the revised version.

Introduction

In my reading, this part has improved significantly. The hypothesis are also clear.

Sample

Attrition analysis is relevant because I though the data collection included at least 2 waves. Since authors now mention the study is conducted in cross-sectional setting, there is no need to do attrition analysis. For author’s information: attrition analysis is about the missing observations, not missing variables.

Measures

I would include the Cronbach’s alpha of depression scale as well and since depression is the dependent variable, including whether this scale is a standardized scale (perhaps literature?).

In my understanding, the study only used non-weighted data. Weighted cannot be simply understood = mean/SD and non-weighted=count. One also cannot say that sample N=45.00 from weighted N=300.000 population. Weighted should be calculated if there is/are reason(s) to believe, that the data is no longer representative for the population if the weighted is not included (such as significant drop-out rate in specific group, so the data should be weighted based on that specific group).

Result

The title of Table 2 is presented before Table 1. Please make a correction.

In the data analysis the authors stated that Pearson’s χ2 tests and t-tests were performed to obtain descriptive statistics. However, these specific results were not presented neither in the Table 1 (Descriptive results) nor in text.

The use of standardized coefficient are better for some people to understand the regression results. Many psychologists are used to that type of coefficient. Since the topic is rather psychological than sociological topic, I thought it would be useful to present what the readers are used to. However, it is only personal preference. You can opt to present the non-standardized coefficients.

Since the dependent variable has an ordinal scale, I don’t think the authors can use the linear regression. Rather logit regression should be used and the interpretation should be based on the logit rationale. Some authors opted to treat their ordinal dependent variable as continuous for practical reason. I don’t think that the best choice to make.

I would report the variance explained (R2) for each model, so that one can conclude how many variance is additionally explained by specific model (e.g., family function or environment).

Discussion

With the results of R2, authors could explain more about which level (family, environment) affect the depression more.

Author Response

(The authors gave the same response as above.)

Reviewer 3 Report

Thanks for submitting the revised manuscript. I am glad to see that you have made the changes.

Author Response

Thank you for your valuable review.

Round 2

Reviewer 2 Report

Dear Authors,

Thank you for the modification.

This manuscript is a resubmission of an earlier submission. The following is a list of the peer review reports and author responses from that submission.

Round 1

Reviewer 1 Report

Need to discuss flourishing with a iterature review. The study method was reasonable but the measures/scales used need to be   named with references. Also need to note that the youth were not interviewed - only parents in the method section. It would also be useful to identify the country in which the study was carried out - rather than leaving it up to the reader to assume. 

Several other critical issueds were noted. One, the references were largely old- more exploration is needed of recent literature. Two, the sentence structure throughout the manuscript needs review and revision. 

Reviewer 2 Report

1. There are English language issues. Please have a native English speaker review article.

2. The first sentence needs to be rewritten into active voice, review line 29.

3. Define the term flourishing in the beginning linking to data, for clarity. The article appears to be written about depression, which is included in the data not flourishing, which is not mentioned till line 97.

4. There are two tables-a descriptive table and a Table identifying Depression. I would expect a table clearly addressing "Flourishing".

5. Reference research by referring to author(s) last names throughout not just reference number.

Reviewer 3 Report

This manuscript wants to examine the relationship between children’s flourishing and mental health (depression). However, I cannot recommend this manuscript for publication since there are substantial issues related to this study.

My biggest concern is about the structure and concept of the study that require significant modification. The hypothesis, results and discussion parts are not coherent. I think it is very hard for the readers to understand the relationship between these mentioned parts. Therefore, I would suggest that authors specify their research questions and conduct analysis that coherence with these questions. Maybe the authors can state their motivation to investigate children’s and parents’ characteristic as predictors for children’s mental health with environment factors a covariates.   

Introduction

  1. Too much explanation about depression, its symptoms and risk factors, which actually have nothing to do with the hypothesis.
  2. What is the definition of children’s flourishing and explain more how this construct can decrease depression, affect a positive mental health etc.
  3. Line 56-57: Both parents and teachers are key role models for children. However, teachers was only mentioned once in this manuscript. Why should authors mention it in the first place?
  4. I would suggest to rewrite the introduction part, shorten the first 4 paragraphs and describing more about children’s flourishing and its relationship with all measured variables in the present study.

Sample

  1. Explain sample for this specific study, mean and standard deviation of age and sex of children (it is mentioned in abstract, however, without standard deviation of age). I would recommend to write them again in the sample part.
  2. Please specify the attrition analysis which shows the pattern of missings values across the waves, for example towards socio-economic status and perhaps gender.

Measures

  1. It is hard to follow this part since everything mixed. Please specify every single measure in at least one paragraph and explain everything about this measure. For example, authors started to explain about pediatric depression that has 4 likert scale. Without explaining this construct more, authors moved on and explained the children’s flourishing.  
  2. Children’s behavior was introduced in the measurement part for the first time. This construct was not even mentioned in the introduction. Please mention this variable also in the introduction!
  3. I would suggest to differentiate the measures into 4 parts: children, parents, family and environment.
  4. Please specify the attrition analysis which shows the pattern of missings values across the waves, for example towards socio-economic status and perhaps gender.

Results

  1. I sense a misunderstanding about weights here. Weighted data is usually used if some cases have different weights compared to others. The aim is to make data more representative for the whole population. If the authors used weighted data, would be nice if the weighted procedure was also shortly described.
  2. It appears to me that authors do not only investigate about children’s flourishing but also all variables including parents, family and environment. Children’s flourishing did not get any special focus of analysis. In this way, there is little coherence between hypothesis and results. I think this is a substantial problems authors need to deal with.
  3. I would prefer to use standardized beta coefficient.
  4. I don’t think it is necessary to include CI in your step-wise regression.

Discussion

  1. Discussion part also reflects that authors have focused their investigation on all variables measured including parental characteristics and environment. This is inconsistent with their hypothesis about the effect of children’s flourishing on depression with several covariates implemented in the model.
  2. Measures related to schools are bullying and school missing. However, the authors made inferences about teacher’s role (line 270). I think one cannot make any interpretation about teachers if he/she did not have any data about them (related to them).
  3. Overall, I suggest the authors to rethink what they want to investigate, specify it in a hypothesis (or several hypothesis) and examine the data based on this concept. Afterwards, the discussion part should also be rewrite according to the finding and its relationship with research questions/hypothesis.

Reviewer 4 Report

The term "flourisihing" will create confusion among readers. Generally the term used for children development is "growth". Not sure why the authors used "flourish" here.

Line 15, data used for this is almost 10 years old. Findings from this data will not make any difference now. There are so many published research of similar types so what this research is offering.

Line 29, is the research focusing on depression or overall mental health? 

Line 32, the authors are trying to explore depression here  and showing depression related health issues.

Line 47 and 53 clearly states that this research is about depression, not on mental health issues most important, there is no solid link shown with children growth and mental development here.

Line 74, the rationale for conducting the research is very weak, fails to show a clear argument and the need for this research.

Line 92, obesity data collected from 10 to 17 years of children. Are they children or adolescents? Also, 0 to 17 is considered as here children not adolescents. This is not clear.

Line 93 does not make any sense

No information on sampling technique and sample size, it shows that the research is heavily flawed and not ready for publication at all.